# Langevin Dynamics with Continuous Tempering for Training Deep Neural Networks

**Nanyang Ye**
University of Cambridge
Cambridge, United Kingdom
yn272@cam.ac.uk

**Zhanxing Zhu**
Center for Data Science, Peking University
Beijing Institute of Big Data Research (BIBDR)
Beijing, China
zhanxing.zhu@pku.edu.cn

**Rafal K.Mantiuk**
University of Cambridge
Cambridge, United Kingdom
rafal.mantiuk@cl.cam.ac.uk

## Abstract

Minimizing non-convex and high-dimensional objective functions is challenging, especially when training modern deep neural networks. In this paper, a novel approach is proposed which divides the training process into two consecutive phases to obtain better generalization performance: Bayesian sampling and stochastic optimization. The first phase is to explore the energy landscape and to capture the 'fat" modes; and the second one is to fine-tune the parameter learned from the first phase. In the Bayesian learning phase, we apply continuous tempering and stochastic approximation into the Langevin dynamics to create an efficient and effective sampler, in which the temperature is adjusted automatically according to the designed "temperature dynamics". These strategies can overcome the challenge of early trapping into bad local minima and have achieved remarkable improvements in various types of neural networks as shown in our theoretical analysis and empirical experiments.

## 1 Introduction

Minimizing non-convex error functions over continuous and high-dimensional spaces has been a primary challenge. Specifically, training modern deep neural networks presents severe difficulties, mainly because of the large number of critical points with respect to the number of dimensions, including various saddle points and local minima [9, 5]. In addition, the landscapes of the error functions are theoretically and computationally impossible to characterize rigidly.

Recently, some researchers have attempted to investigate the landscapes of the objective functions for several types of neural networks. Under some strong assumptions, previous works [21, 4, 12] showed that there exists multiple, almost equivalent local minima for deep neural networks, using a wide variety of theoretical analysis and empirical observations. Despite of the nearly equivalent local minima during training, obtaining good generalization performance is often more challenging with current stochastic gradient descent (SGD) or some of its variants. It was demonstrated in [22] that deep network structures are sensitive to initialization and learning rates. And even networks without nonlinear activation functions may have degenerate or hard to escape saddle points [12].

One important reason of the difficulty to achieve good generalization is, that SGD and some variants may tend to trap into a certain local minima or flat regions with poor generalization property [25, 1, 13].

In other words, most of existing optimization methods do not explore the landscapes of the error functions efficiently and effectively. To increase the possibility of sufficient exploration of the parameter space, [25] proposed to train multiple deep networks in parallel and made individual networks explore by modulating their distance to the ensemble average.

Another kind of approaches attempt to tackle this issue through borrowing the idea of classical simulated annealing or tempering [15, 6, 10]. The authors of [19] proposed to inject Gaussian noise with annealed variance (corresponding to the annealed temperature in simulated annealing) into the standard SGD to make the original optimization dynamics more "stochastic". In essence, this approach is the same as a scalable Bayesian learning method, Stochastic Gradient Langevin Dynamics (SGLD [24]) with decayed stepsizes. The Santa algorithm [1] incorporated a similar idea into a more sophisticated stochastic gradient Markov Chain Monte Carlo (SG-MCMC) framework. However, previous studies show that the efficiency and performance of these methods for training deep neural networks is very sensitive to the annealing schedule of the temperature in these methods. Slow annealing will lead to significantly slow optimization process as observed in the literature of simulated annealing [10], while fast annealing hinders the exploration dramatically, leading to the optimizer trapped in poor local minima too early. Unfortunately, searching for a suitable annealing schedule for training deep neural network is hard and time-consuming according to empirical observations in these works.

To facilitate more efficient and effective exploration for training deep networks, we divide the whole training process into two phases: Bayesian sampling for exploration and optimization for fine-tuning. The motivation of implementing a sampling phase is that sampling is theoretically capable of fully exploring the parameter space and can provide a good initialization for optimization phase. This strategy is motivated by the sharp minima theory [13] and its validity will be verified by our empirical experiments.

Crucially, in the sampling phase, we employ the idea of continuous tempering [8, 17] in molecule dynamics [20], and implement an extended stochastic gradient second-order Langevin dynamics *with smoothly varying temperatures*. Importantly, the change of temperature is governed *automatically* by a specifically designed dynamics coupled with the original Langevin dynamics. This is different from the idea of simulated annealing adopted in [19, 1], in which the temperature is only allowed to decrease according to a manually predefined schedule. Our "temperature dynamics" is beneficial in the sense that it increases the capability of exploring the energy landscapes and hopping between different modes of the sampling distributions. Thus, it may avoid the problem of early trapping into bad local minima that exists in other algorithms. We name our approach **CTLD**, abbreviated for "Continuously Tempered Langevin Dynamics". With support of extensive empirical evidence, we demonstrated the efficiency and effectiveness of our proposed algorithm for training various types of deep neural networks. To the best of our knowledge, this is the first attempt that adopts continuous tempering into training modern deep networks and produces remarkable improvements over the state-of-the-art techniques.

## 2 Preliminaries

The goal of training deep neural network is to minimize the objective function $U(\boldsymbol{\theta})$ corresponding to a non-convex model of interest, where $\boldsymbol{\theta} \in \mathbb{R}^d$ are the model parameters. In a Bayesian setting, the objective $U(\boldsymbol{\theta})$ can be treated as the potential energy function, i.e., the negative log posterior, $U(\boldsymbol{\theta}) = -\sum_{i=1}^{N} \log \mathbb{p}(\mathbf{x}_i | \boldsymbol{\theta}) - \log \mathbb{p}_0(\boldsymbol{\theta})$, where $\mathbf{x}_i$ represents the $i$-th observed data point, $\mathbb{p}_0(\boldsymbol{\theta})$ is the prior distribution for the model parameters and $\mathbb{p}(\mathbf{x}_i | \boldsymbol{\theta})$ is the likelihood term for each observation. In optimization scenario, the counterpart of the complete negative log likelihood is the loss function and $-\log \mathbb{p}_0(\boldsymbol{\theta})$ is typically referred to as a regularization term.

### 2.1 Stochastic Gradient MCMC

In the scenario of Bayesian learning, obtaining the samples of a high-dimensional distribution is a necessary procedure for many tasks. Classic dynamics offers such a way to sample the distribution.

The Hamiltonian in classic dynamics is $H(\boldsymbol{\theta}, \mathbf{r}) = U(\boldsymbol{\theta}) + \frac{1}{2}\mathbf{r}^T\mathbf{r}$, the sum of the potential energy $U(\boldsymbol{\theta})$ and kinetic energy $\frac{1}{2}\mathbf{r}^T\mathbf{r}$, where $\mathbf{r} \in \mathbb{R}^d$ is the momentum term Standard (second-order) Langevin dy-

namics[1] with constant temperature $T_c$ can be described by following stochastic differential equations (SDEs),

$$d\boldsymbol{\theta} = \mathbf{r}dt, \quad d\mathbf{r} = -\nabla_{\boldsymbol{\theta}} U(\boldsymbol{\theta})dt - \gamma\mathbf{r}dt + \sqrt{2\gamma\beta^{-1}}d\mathbf{W} \quad (1)$$

where $\nabla_{\boldsymbol{\theta}} U(\boldsymbol{\theta})$ is the gradient of the potential energy w.r.t. the configuration states $\boldsymbol{\theta}$, $\gamma$ denotes the friction coefficient, $\beta^{-1} = k_B T_c$ with Boltzmann constant $k_B$, and $d\mathbf{W}$ is the standard Wiener process. In the context of this work for Markov Chain Monte Carlo (MCMC) and optimization theory, we always assume $\beta = 1$ for simplicity.

If we simulate the dynamics in Eqs (1), a well-known stationary distribution can be achieved [20], $\mathbb{p}(\boldsymbol{\theta}, \mathbf{r}) = \exp\left(-\beta H(\boldsymbol{\theta}, \mathbf{r})\right)/Z$, where $Z = \int\int \exp\left(-\beta H(\boldsymbol{\theta}, \mathbf{r})\right) d\boldsymbol{\theta}d\mathbf{r}$ is the normalization constant for the probability density. The desired probability distribution associated with the parameters $\boldsymbol{\theta}$ can be obtained by marginalizing the joint distribution, $\mathbb{p}(\boldsymbol{\theta}) = \int \mathbb{p}(\boldsymbol{\theta}, \mathbf{r})d\mathbf{r} \propto \exp\left(-\beta U(\boldsymbol{\theta})\right)$. The MCMC procedures using the analogy of dynamics described by SDEs are often referred to as dynamics-based MCMC methods.

However, in the "Big Data" settings with large $N$, evaluating the full gradient term $\nabla_{\boldsymbol{\theta}} U(\boldsymbol{\theta})$ is computationally expensive. The usage of stochastic approximation reduces the computational burden dramatically, where a much smaller subset of the data, $\{\mathbf{x}_{k_1}, \ldots, \mathbf{x}_{k_m}\}$, is selected randomly to approximate the full one,

$$\tilde{U}(\boldsymbol{\theta}) = -\frac{N}{m}\sum_{j=1}^{m} \log \mathbb{p}(\mathbf{x}_{k_j}|\boldsymbol{\theta}) - \log \mathbb{p}_0(\boldsymbol{\theta}). \quad (2)$$

And the resulting stochastic gradient $\nabla\tilde{U}(\boldsymbol{\theta})$ is an unbiased estimation of the true gradient. Then the stochastic gradient approximation can be used in the dynamics-based MCMC methods, often referred to as SG-MCMC, such as [24, 3].

## 2.2 Simulated Annealing for Global Optimization

Simulated annealing (SA [15, 6, 10]) is a probabilistic technique for approximating the global optimum of a given function $U(\boldsymbol{\theta})$. A Brownian-type of diffusion algorithm was proposed [6] for continuous optimization by discretizing the following SDE,

$$d\boldsymbol{\theta} = -\nabla U(\boldsymbol{\theta})dt + \sqrt{2\beta^{-1}(t)}d\mathbf{W}, \quad (3)$$

where $\beta^{-1}(t) = k_B T(t)$ decays as $T(t) = c/\log(2 + t)$ with a sufficiently large constant $c$, to ensure theoretical convergence. Unfortunately, this logarithmic annealing schedule is extremely slow for optimization. In practice, the polynomial schedules are often adopted to accelerate the optimization processes though without any theoretical guarantee, such as $T(t) = c/(a + t)^b$, where $a > 0, b \in (0.5, 1), c > 0$ are hyperparameters. Recently, [19, 1] incorporated the simulated annealing with this polynomial cooling schedule into the training of neural networks. A critical issue behind these methods is that the generalization performance and efficiency of the optimization are highly sensitive to the cooling schedule. Unfortunately, searching for a suitable annealing schedule for training deep neural network is hard and time-consuming according to empirical observations in these works.

These challenges motivate our work. We proposed to divide the whole optimization process into two phases: Bayesian sampling based on stochastic gradient for parameter space exploration and standard SGD with momentum for parameters optimization. The key step in the first phase is that we employ a new tempering scheme to facilitate more effective exploration over the whole energy landscape. Now, we will elaborate on the proposed approach.

# 3 Two Phases for Training Neural Networks

As mentioned in Section 1, the objective functions of deep networks contain multiple, nearly equivalent local minima. The key difference between these local minima is whether they are "*flat*" or "*sharp*", i.e., lying in "wide valleys" or "stiff valleys". A recent study by [13] showed that sharp

minima often lead to poorer generalization performance. Flat minimizers of the energy landscape tend to generalize better due to their robustness to data perturbations, noise in the activations as well as perturbations of the parameters. However, most of existing optimization methods lack the ability to efficiently explore the flat minima, often trapping into sharp minima too early.

We consider this issue in a Bayesian way: the flat minima corresponds to "fat" modes of the induced probability distribution over $\boldsymbol{\theta}$, $\mathbb{p}(\boldsymbol{\theta}) \propto \exp\left(-U(\boldsymbol{\theta})\right)$. Obviously, these fat modes own much more probability mass than "thin" ones since they are nearly as "tall" as each other. Based on this simple observation, we propose to implement a Bayesian sampling procedure before the optimization phase. Bayesian learning is capable of exploring the energy landscape more thoroughly. Due to the large probability mass, the sampler tends to capture the desired regions near the "flat" minima. This provides a good starting region for optimization phase to fine-tune the parameters learning.

When sampling the distribution $\mathbb{p}(\boldsymbol{\theta})$, the multi-modality issue demands the samplers to transit between isolated modes efficiently. To this end, we incorporate the continuous tempering and stochastic approximation techniques into the Langevin dynamics to derive an efficient and effective sampling process for training deep neural networks.

## 4 CTLD: Continuously Tempered Langevin Dynamics

Faced with high-dimensional and non-convex energy landscapes $U(\boldsymbol{\theta})$, such as the error functions in deep neural networks, the key challenge is how to efficiency and effectively explore the energy landscapes. Inspired by the idea of continuous tempering [8, 17] in molecule dynamics, we incorporate the "temperature dynamics" and stochastic approximation into the Langevin dynamics in a principled way to allow a more effective exploration of the energy landscape. The temperature in CTLD evolves *automatically* governed by the embedded "temperature dynamics", which is different from the predefined annealing schedules used in [19, 1].

The primary dynamics we use for Bayesian sampling is as follows,

$$
\begin{aligned}
\mathrm{d}\boldsymbol{\theta} &= \mathbf{r}\mathrm{d}t, \quad \mathrm{d}\mathbf{r} = -\nabla_{\boldsymbol{\theta}} U(\boldsymbol{\theta})\mathrm{d}t - \gamma\mathbf{r}\mathrm{d}t + \sqrt{2\gamma\tilde{\beta}^{-1}(\alpha)}\mathrm{d}\mathbf{W} \\
\mathrm{d}\alpha &= r_\alpha \mathrm{d}t, \quad \mathrm{d}r_\alpha = h(\boldsymbol{\theta}, \mathbf{r}, \alpha)\mathrm{d}t - \gamma_\alpha r_\alpha \mathrm{d}t + \sqrt{2\gamma_\alpha}\mathrm{d}W_\alpha,
\end{aligned}
\tag{4}
$$

where $\alpha$ is the newly augmented variable to control the inverse temperature $\tilde{\beta}$, $\gamma_\alpha$ is the corresponding friction coefficient. Note that $\tilde{\beta}^{-1}(\alpha) = k_B T(\alpha) = 1/g(\alpha)$, depending on the augmented variable $\alpha$ to dynamically adjust the temperature. The function $g(\alpha)$ plays the role as scaling the constant temperature $T_c$. The dynamics of $\boldsymbol{\theta}$ and $\alpha$ are coupled through the function $h(\boldsymbol{\theta}, \mathbf{r}, \alpha)$. Both of the two functions will be described later.

It can be shown that if we simulate the SDEs described in Eq (4), the following stationary distribution will be achieved [8],

$$
\mathbb{p}(\boldsymbol{\theta}, \mathbf{r}, \alpha, r_\alpha) \propto \exp\left(-H_e(\boldsymbol{\theta}, \mathbf{r}, \alpha, r_\alpha)\right),
\tag{5}
$$

with the extended Hamiltonian and the coupling function $h(\cdot)$ as

$$
H_e(\boldsymbol{\theta}, \mathbf{r}, \alpha, r_\alpha) = g(\alpha)H(\boldsymbol{\theta}, \mathbf{r}) + \phi(\alpha) + r_\alpha^2/2, \quad h(\boldsymbol{\theta}, \mathbf{r}, \alpha) = -\partial_\alpha g(\alpha)H(\boldsymbol{\theta}, \mathbf{r}) - \partial_\alpha \phi(\alpha), \tag{6}
$$

where $\phi(\alpha)$ is some confining potential to enforce additional properties of $\alpha$, discussed in Section 4.2. The proof of achievement of this stationary distribution $\mathbb{p}(\boldsymbol{\theta}, \mathbf{r}, \alpha, r_\alpha)$ is provided in the Supplementary Material for completeness.

In order to allow the system to overcome the issue of muli-modality efficiently, the temperature scaling function $g(\alpha)$ can be any convenient form that satisfies: $0 < g(\alpha) \leq 1$ and being smooth. This will allow the system to experience different temperature configurations smoothly. A simple choice would be the following piecewise polynomial function, with $z(\alpha) = \frac{|\alpha|-\delta}{\delta'-\delta}$,

$$
g(\alpha) = \begin{cases} 1, & \text{if } |\alpha| \leq \delta, \\ 1 - S\left(3z^2(\alpha) - 2z^3(\alpha)\right), & \text{if } \delta < |\alpha| < \delta' \\ 1 - S, & \text{if } |\alpha| \geq \delta', \end{cases}
\tag{7}
$$

Figure 1 presents this temperature scaling function with $\delta = 0.4, \delta' = 1.5$ and $S = 0.85$. In this case, $\tilde{\beta}^{-1}(\alpha) \in [1 - S, 1]$. Experiencing high temperature configurations continuously allows the

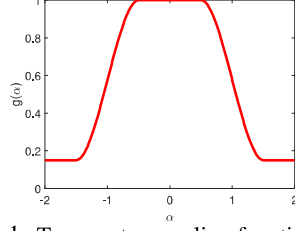

Figure 1: Temperature scaling function $g(\alpha)$.

sampler to explore the parameter space more "wildly", significantly alleviating the issue of trapping into local minima or flat regions. Moreover, it can be easily seen that when $g(\alpha) = 1$, we can recover the desired distribution $\mathbb{p}(\boldsymbol{\theta}) \propto \exp(-U(\boldsymbol{\theta}))$.

## 4.1 Stochastic Approximation for CTLD

With large-scale datasets, we adopt the technique of stochastic approximation to estimate the full potential term $U(\theta)$ and its gradient $\nabla U(\boldsymbol{\theta})$, as shown in Eq. (2). One way to analyse the impact of the stochastic approximation is to make use of the central limit theorem,

$$\tilde{U}(\boldsymbol{\theta}) = U(\boldsymbol{\theta}) + \mathcal{N}\left(0, \sigma^2(\boldsymbol{\theta})\right), \quad \nabla_{\boldsymbol{\theta}}\tilde{U}(\boldsymbol{\theta}) = \nabla_{\boldsymbol{\theta}}U(\boldsymbol{\theta}) + \mathcal{N}\left(\mathbf{0}, \boldsymbol{\Sigma}(\boldsymbol{\theta})\right) \tag{8}$$

The usage of stochastic approximation results in a noisy potential term and gradient. Simply plugging in the the noisy estimation into the original dynamics will lead to a dynamics with additional noise terms. To dissipate the introduced noise, we assume the covariance matrices, $\sigma^2(\boldsymbol{\theta})$ and $\boldsymbol{\Sigma}(\boldsymbol{\theta})$, are available, and satisfy the positive semi-definiteness, $2\gamma\tilde{\beta}^{-1}(\alpha)\mathbf{I} - \eta\boldsymbol{\Sigma}(\boldsymbol{\theta}) \succcurlyeq \mathbf{0}$ and $2\gamma_\alpha - \eta\partial_\alpha g(\alpha)\sigma^2(\boldsymbol{\theta}) \geq 0$ with $\eta$ as the associated step size of numerical integration for the SDEs. With $\eta$ small enough, this is always true since the introduced stochastic noise scales down faster than the added noise. Then, we propose CTLD with stochastic approximation,

$$\begin{aligned} \mathrm{d}\boldsymbol{\theta} &= \mathbf{r}\mathrm{d}t, \quad \mathrm{d}\mathbf{r} = -\nabla_{\boldsymbol{\theta}}\tilde{U}(\boldsymbol{\theta})\mathrm{d}t - \gamma\mathbf{r}\mathrm{d}t + \sqrt{2\gamma\tilde{\beta}^{-1}(\alpha)\mathbf{I} - \eta\boldsymbol{\Sigma}(\boldsymbol{\theta})}\mathrm{d}\mathbf{W} \\ \mathrm{d}\alpha &= r_\alpha\mathrm{d}t, \quad \mathrm{d}r_\alpha = \tilde{h}(\boldsymbol{\theta}, \mathbf{r}, \alpha)\mathrm{d}t - \gamma_\alpha r_\alpha\mathrm{d}t + \sqrt{2\gamma_\alpha - \eta\partial_\alpha g(\alpha)\sigma^2(\boldsymbol{\theta})}\mathrm{d}W_\alpha, \end{aligned} \tag{9}$$

where the coupling function $\tilde{h}(\boldsymbol{\theta}, \mathbf{r}, \alpha) = -\partial_\alpha g(\alpha)\left(\tilde{U}(\boldsymbol{\theta}) + \mathbf{r}^T\mathbf{r}/2\right) - \partial_\alpha\phi(\alpha)$. Then the following theorem to show the stationary distribution of the dynamics described in Eq. (9).

**Theorem 1.** $\mathbb{p}(\boldsymbol{\theta}, \mathbf{r}, \alpha, r_\alpha) \propto \exp\left(-H_e(\boldsymbol{\theta}, \mathbf{r}, \alpha, r_\alpha)\right)$ *is the stationary distribution of the dynamics SDEs Eq. (9), when the variance terms* $\sigma^2(\boldsymbol{\theta})$ *and* $\boldsymbol{\Sigma}(\boldsymbol{\theta})$ *are available.*

The proof for this theorem is provided in the Supplementary Materials. In practical implementation of simulating the $\mathbf{r}$ and $r_\alpha$ of Eq. (9), we have

$$\mathbf{r}^{(t)} = (1 - \eta^{(t)}\gamma)\mathbf{r}^{(t-1)} - \nabla_{\boldsymbol{\theta}}\tilde{U}(\boldsymbol{\theta}^{(t)})\eta^{(t)} + \mathcal{N}\left(\mathbf{0}, \frac{2\eta^{(t)}\gamma}{g(\alpha^{(t-1)})}\mathbf{I} - (\eta^{(t)})^2\hat{\boldsymbol{\Sigma}}(\boldsymbol{\theta}^{(t-1)})\right) \tag{10}$$

$$r_\alpha^{(t)} = (1 - \eta^{(t)}\gamma_\alpha)r_\alpha^{(t-1)} + \tilde{h}(\boldsymbol{\theta}^{(t)}, \mathbf{r}^{(t)}, \alpha^{(t)})\eta^{(t)} + \mathcal{N}(0, 2\eta^{(t)}\gamma_\alpha - (\eta^{(t)})^2\hat{\sigma}^2(\boldsymbol{\theta})),$$

where $\hat{\boldsymbol{\Sigma}}(\boldsymbol{\theta})$ and $\hat{\sigma}^2(\boldsymbol{\theta})$ are the estimation of the noise variance terms. In Eq. (10), the noise introduced by the stochastic approximation is compensated by multiplying $(\eta^{(t)})^2$. To avoid the estimation of the variance terms, we often choose $\eta^{(t)} = \eta$ small enough and $\gamma, \gamma_\alpha$ large enough to make the $\eta^2\hat{\boldsymbol{\Sigma}}(\boldsymbol{\theta})$ and $\eta^2\hat{\sigma}^2(\boldsymbol{\theta})$ numerically negligible, and thus ignored in practical use.

## 4.2 Control of The Augmented Variable

It is expected that the distribution of experienced temperatures of the system should only depend on the form of the scaling function $g(\alpha)$. This would help us achieve the desired temperature distribution, thus resulting in a more controllable system. To this end, two strategies are shown in this part.

Firstly, we confine the augmented variable $\alpha$ to be in the interval $[-\delta', \delta']$. One simple choice to achieve this is to configure its gradient as a "force well":

$$\partial_\alpha\phi(\alpha) = \begin{cases} 0, & \text{if } |\alpha| \leq \delta' \\ C, & \text{otherwise,} \end{cases} \tag{11}$$

where $C$ is some appropriate constant. Intuitively, when the particle $\alpha$ "escapes" from the interval $[-\delta', \delta']$, a force induced by $\partial_\alpha \phi(\alpha)$ will "pull" it back.

Secondly, we restrict the distribution of $\alpha$ to be *uniform* over the specified range. Together with the design of $g(\alpha)$, this restriction can guarantee the required percent of running time for sampling with the original inverse temperature $\beta = 1$, and the remaining for high temperatures. For example, in case of $g(\alpha)$ in Eq.(7), the percent of simulation time for high temperatures is $(1 - \delta/\delta')100\%$.

An adaptive biasing method metadynamics [16] can be used to achieve a flat density across a bounded range of $\alpha$. Metadynamics incorporates a history-dependent potential term to gradually fill the minima of energy surface corresponding to $\alpha$'s marginal density, resulting in a uniform distribution of $\alpha$. In essence, metadynamics biases the extended Hamiltonian by an additional potential $V_b(\alpha)$,

$$H_m(\boldsymbol{\theta}, \mathbf{r}, \alpha, r_\alpha) = g(\alpha)H(\boldsymbol{\theta}, \mathbf{r}) + \phi(\alpha) + r_\alpha^2/2 + V_b(\alpha) \tag{12}$$

The bias potential term is initialized $V_b^{(0)}(\alpha) = 0$, and then updated by iteratively adding Gaussian kernel terms,

$$V_b^{(t+1)}(\alpha) = V_b^{(t)}(\alpha) + w \exp\left(-(\alpha - \alpha^{(t)})^2/(2\sigma^2)\right), \tag{13}$$

where $\alpha^{(t)}$ is the value of the $t$-th time step during simulation, the magnitude $w$ and variance term $\sigma^2$ are hyperparameters. To update the bias potential over the range $[-\delta', \delta]$, we can discretize this interval into $K$ equal bins, $\{-\delta', \alpha_1^{(t)}, \ldots, \alpha_{K-1}^{(t)}, \delta'\}$ and in each time step update $\alpha$ in each bin. Thus, the force induced by the bias potential can be approximated by the difference between adjacent bins divided by the length of each bin. The force $h(\boldsymbol{\theta}^{(t)}, \mathbf{r}^{(t)}, \alpha^t)$ over the particle $\alpha$ will be biased due to the force induced by metadynamics,

$$\tilde{h}(\boldsymbol{\theta}^{(t)}, \mathbf{r}^{(t)}, \alpha^{(t)}) \leftarrow \tilde{h}(\boldsymbol{\theta}^{(t)}, \mathbf{r}^{(t)}, \alpha^{(t)}) - \frac{V_b^{(t)}(\alpha_{k^*+1}) - V_b^{(t)}(\alpha_{k^*})}{2\delta'/K} \tag{14}$$

where $k^*$ denotes the bin index inside which $\alpha^{(t)}$ is located. Finally, we summarize CTLD in Alg. 1.

---

### Algorithm 1: Continuously Tempered Langevin Dynamics

**Input:** $m, \eta$, number of steps for sampling $L_s$, $\gamma, \gamma_\alpha$; metadynamics parameters: $C, w, \sigma^2$ and $K$.
Initialize $\boldsymbol{\theta}^{(0)}, \mathbf{r}^{(0)} \sim \mathcal{N}(\mathbf{0}, \mathbf{I})$, $\alpha^{(0)} = 0, r_\alpha^{(0)} \sim \mathcal{N}(0, 1)$, and $V_b^{(0)}(\alpha^{(0)}) = 0$.
**for** $t = 1, 2, \ldots$ **do**
    Randomly sample a minibatch of the dataset with size $m$ to obtain $\tilde{U}(\boldsymbol{\theta}^{(t)})$;
    **if** $t < L_s$ **then**
        Sample $\boldsymbol{\epsilon} \sim \mathcal{N}(\mathbf{0}, \mathbf{I}), \epsilon_\alpha \sim \mathcal{N}(0, 1)$;
        $\boldsymbol{\theta}^{(t)} = \boldsymbol{\theta}^{(t-1)} + \eta \mathbf{r}^{(t-1)}, \quad \mathbf{r}^{(t)} = (1 - \eta\gamma)\mathbf{r}^{(t-1)} - \nabla_{\boldsymbol{\theta}}\tilde{U}(\boldsymbol{\theta}^{(t)})\eta + \sqrt{\frac{2\eta\gamma}{g(\alpha^{(t-1)})}}\boldsymbol{\epsilon}$
        $\alpha^{(t)} = \alpha^{(t-1)} + \eta r_\alpha^{(t-1)}$.
        Update $V_b(\alpha)$ according to Eq. (13); Find the $k^*$ indexing which bin $\alpha^{(t)}$ is located in.
        $\tilde{h}(\boldsymbol{\theta}^{(t)}, \mathbf{r}^{(t)}, \alpha^{(t)}) = -\partial_\alpha g(\alpha^{(t)})\tilde{H}(\boldsymbol{\theta}^{(t)}, \mathbf{r}^{(t)}) - \partial_\alpha\phi(\alpha^{(t)}) - \frac{V_b^{(t)}(\alpha_{k^*+1}) - V_b^{(t)}(\alpha_{k^*})}{2\delta'/K}$
        $r_\alpha^{(t)} = (1 - \eta\gamma_\alpha)r_\alpha^{(t-1)} + \tilde{h}(\boldsymbol{\theta}^{(t)}, \mathbf{r}^{(t)}, \alpha^{(t)})\eta + \sqrt{2\eta\gamma_\alpha}\epsilon_\alpha$
    **else**
        $\boldsymbol{\theta}^{(t)} = \boldsymbol{\theta}^{(t-1)} + \eta\mathbf{r}^{(t-1)}, \quad \mathbf{r}^{(t)} = (1 - \eta\gamma)\mathbf{r}^{(t-1)} - \nabla_{\boldsymbol{\theta}}\tilde{U}(\boldsymbol{\theta}^{(t)})\eta$
    **end if**
**end for**

---

### 4.3 Connection with Other Methods

There is a direct relationship between the proposed method and SGD with momentum. In the optimization phase, CTLD essentially implements SGD with momentum: as shown in SGHMC [3], the learning rate in the SGD with momentum corresponds to $\eta^2$ in our method, the momentum coefficient the SGD is equivalent to $1 - \eta\gamma$. The key difference appears in the Bayesian learning phase, a dynamical diffusion term $\sqrt{\frac{2\eta\gamma}{g(\alpha^{(t-1)})}}\boldsymbol{\epsilon}$ is added to the update of the momentum to empower the sampler/optimizer to explore the parameter space more thoroughly. This directly avoids the issue of being stuck into poor local minima too early. CTLD introduces stochastic approximation and temperature dynamics into the Langevin dynamics in a principled way. This distinguishes it from the deterministic annealing schedules adopted in Santa [1] and SGLD/AnnealSGD [24, 19].

## 4.4 Parameter Settings, Computational Time and Convergence Analysis

Though there exists several hyperparameters in our method, in practice, the main parameters we need to tune are the learning rate and the momentum (i.e. related to friction coefficients). Across all the experiments, for other hyperparamters, including those in confining potential function $\phi(\alpha)$ and metadynamics, we fix them with our empirical formulae relying on the learning rate. See Supplementary Materials for a thorough analysis on hyperparameter settings. Moreover, through our sensitivity analysis for these hyperparameters, we find they are quite robust to algorithm performance within our estimation range, as shown in Section 5.3. Therefore, practical users can just tune the learning rate and momentum to use CTLD for training neural networks, which is as simple as SGD with momentum.

Compared with SGD with momentum, our proposal CTLD only introduces an additional 1D augmented variable $\alpha$, and its computational cost in almost negligible, as shown in Supplementary Materials. The convergence analysis of CTLD is also provided in the Supplementary Materials to demonstrate its stability.

## 5 Experiments

To evaluate the proposed method, we conduct experiments on stacked denoising autoencoders and character-level recurrent neural networks. The comparing methods include SGD with momentum, RMSprop, Adam [14], AnnealSGD [19], Santa [1] and our proposal CTLD. The same parameter initialization method "Xavier" [7] is used except for character recurrent neural networks. The hyperparameter settings for each compared method are implemented by grid search, provided in the Supplementary Materials.

### 5.1 Stacked Denoising Autoencoders

Stacked denoising autoencoders (SdA) [23] have been proven to be useful in pre-training neural networks for improved performance. We focus on the greedy layer-wise training procedure of SdAs. Dropout layers are appended to each layer with a rate of $0.2$ except for the first and last layer. We use the training set of MNIST data consisting of 60000 training images for this task. The network is fully connected, 784-500-500-2000-10. The learning curves of mean square errors (MSE) for each method are shown in Figure. 2(a). The bumps in iteration $1, 2, 3 \times 10^5$ is due to the switching to next layer during training. Though CTLD in the sampling phase is not as fast as other methods, it can find the regions of good minima, and fine-tune to the best results in final stage.

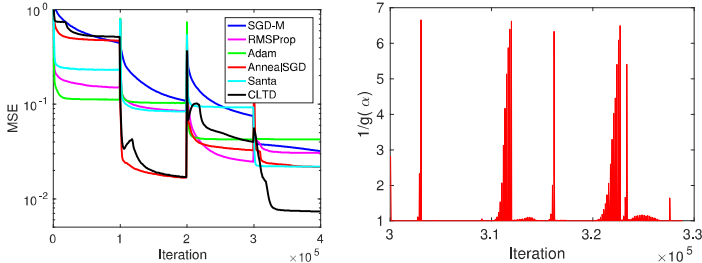

Figure 2: (**Left**) Learning curves of SdAs; (**Right**) The evolution of the noise magnitude $\tilde{\beta} = 1/g(\alpha)$ during the training the final layer.

We also track the evolution of the augmented variable $\alpha$ and plot the noise magnitude $\tilde{\beta}(\alpha) = 1/g(\alpha)$ during the training the final layer, shown in the right panel of Fig. 2. We can observe that the behavior of the magnitude of the noise term is dramatically different from the predefined decreasing schedule used in Santa and AnnealSGD. The temperature dynamics introduced in CTLD adjusts the noise term adaptively. This helps the system to explore the landscape of the loss function more thoroughly and find the regions of good local minima with a higher probability.

### 5.2 Character Recurrent Neural Networks for Language Modeling

We test our method on the task of character prediction using LSTM networks. The objective is to minimize the per-character perplexity, $\frac{1}{N} \sum_{i=1}^{N} \exp \left( \sum_{t=1}^{T_i} -\log p(\mathbf{x}_t^i | \mathbf{x}_1^i, ..., \mathbf{x}_{t-1}^i; \boldsymbol{\theta}) \right)$, where $\boldsymbol{\theta}$

is a set of parameters for the model, $\mathbf{x}_t^n$ is the observed data and $T_i$ is the length of $i$-th sentence. The hidden units are set as LSTM units. We run the models with different methods on the Wikipedia Hutter Prize 100MB dataset with a setting of 3-layer LSTM, 64 hidden layers, the same with the original paper [11]. The training and test perplexity are shown in Fig. 3.

The best training and test perplexities are reached by our method CTLD, which also has the fastest convergence speed. RMSProp and Adam converge very fast in the early iterations, but they seem to be trapped in some poor local minima.

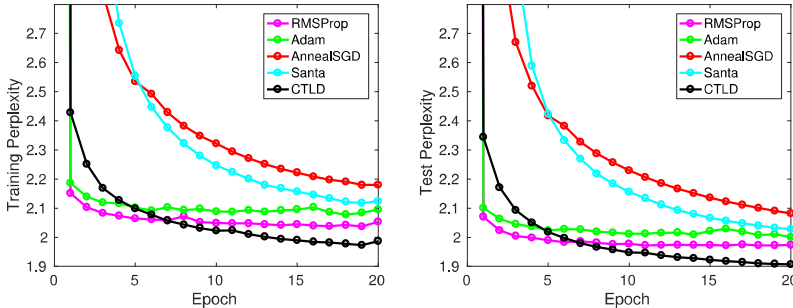

Figure 3: (**Left**) CharRNN on Wiki training set; (**Right**) CharRNN on Wiki test set. Note that SGD-M doest not appear in the Figure because the training and test perplexities for SGD-M are much higher.

### 5.3 Sensitivity Analysis

Since there exist several hyperparameters in CTLD, we analyze the sensitivity of hyperparameter settings within our estimation range (provided in Supplementary Materials). We implement the character-level RNN on *War and Peace* by Leo Tolstoy instead of Wiki dataset, considering the computational speed. The same model architecture is used as [11]. The learning rate is set as $2 \times 10^{-4}$, momentum as $0.7$. We train the model for $50$ epochs until full convergence. The results are shown in Fig. 4. According to Fig. 4, the setting of hyperparameter $w$ and $\sigma$ is robust within our estimation

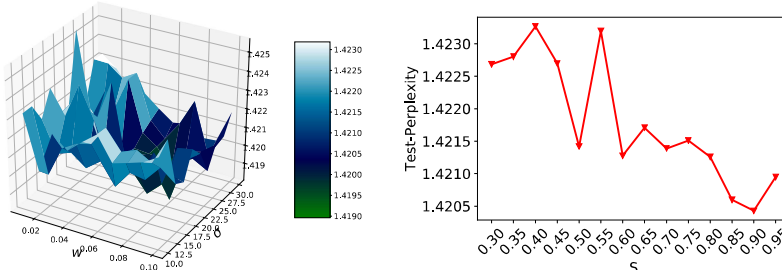

Figure 4: (**Left**) test perplexity versus $w$ and $\sigma$ ($S = 0.85$); (**Right**) test perplexity versus of $S$ ( with $w = 20, \sigma = 0.04$)

range. This also shows metadynamics performs quite stable for training neural networks. For the sensitivity of $S$, with the increase of $S$, the range of temperature enlarges accordingly. Larger range of temperature slightly enhances the ability of CTLD to explore the energy landscape, and leads to better local minima. However, this improvement is quite limited, as shown in Fig. 4, demonstrating the robustness of the hyperparameter $S$. Therefore, we can conclude that with the continuous tempering scheme, our proposed method remains relatively stable under different hyperparameter settings. Practical users only need tune the learning rate and momentum to use CTLD.

## 6 Conclusion & Future Directions

We propose CTLD, an effective and efficient approach for training modern deep neural networks. It involves scalable Bayesian sampling combined with continuous tempering to capture the "fat" modes, and thus avoiding the issue of getting trapped into poor local minima too early. Extensive theoretical and empirical evidence verify the superiority of our proposal over the existing methods. Future directions includes theoretically analyzing the effects of metadynamics and hyperparameter settings, and usage of high-order integrators and preconditioners to improve convergence speed.

## Footnotes

[1]Standard Langevin dynamics is different from that used in SGLD [24], which is the first-order Langevin dynamics, i.e., Brownian dynamics.

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
