[Supplementary Material · supp_revised.pdf]

# A The Proof for Primary Dynamics

*Proof.* We aim to achieve the following stationary distribution through simulating the dynamics in Eq. (4),

$$\mathbb{p}(\boldsymbol{\theta}, \mathbf{r}, \alpha, r_\alpha) \propto \exp\left(-H_e(\boldsymbol{\theta}, \mathbf{r}, \alpha, r_\alpha)\right), \tag{15}$$

with the extended Hamiltonian and the coupling function $h(\cdot)$ as

$$H_e(\boldsymbol{\theta}, \mathbf{r}, \alpha, r_\alpha) = g(\alpha)H(\boldsymbol{\theta}, \mathbf{r}) + \phi(\alpha) + r_\alpha^2/2 \tag{16}$$

$$h(\boldsymbol{\theta}, \mathbf{r}, \alpha) = -\partial_\alpha g(\alpha)H(\boldsymbol{\theta}, \mathbf{r}) - \partial_\alpha \phi(\alpha), \tag{17}$$

Now we derive Fokker-Planck operator of this probability density as follows, where we use $\mathbb{p}$ to represent $\mathbb{p}(\boldsymbol{\theta}, \mathbf{r}, \alpha, r_\alpha)$ for notational simplicity,

$$\begin{aligned}
\mathcal{L}\mathbb{p} &= -\partial_{\boldsymbol{\theta}}(\mathbf{r}\mathbb{p}) + \partial_{\mathbf{r}}\left(\nabla_{\boldsymbol{\theta}}U(\boldsymbol{\theta})\mathbb{p} + \gamma\mathbf{r}\mathbb{p}\right)\mathbb{p} - \partial_\alpha(m_\alpha^{-1}r_\alpha\mathbb{p}) + \partial_{r_\alpha}\left(\gamma_\alpha r_\alpha\mathbb{p} - h(\boldsymbol{\theta}, \mathbf{r}, \alpha)\mathbb{p}\right) \\
&\quad + \partial_{\mathbf{r}}\left(\gamma\tilde{\beta}^{-1}(\alpha)\partial_{\mathbf{r}}\mathbb{p}\right) + \partial_{r_\alpha}\left(\gamma_\alpha \partial_{r_\alpha}\mathbb{p}\right) \\
&= -\partial_{\boldsymbol{\theta}}\partial_{\mathbf{r}}\mathbb{p} + \partial_{\mathbf{r}}\partial_{\boldsymbol{\theta}}\mathbb{p} + \partial_{\mathbf{r}}(\gamma\mathbf{r}\mathbb{p}) - \partial_\alpha(r_\alpha\mathbb{p}) + \partial_{r_\alpha}(\gamma_\alpha r_\alpha\mathbb{p}) - h(\boldsymbol{\theta}, \mathbf{r}, \alpha)\partial_{r_\alpha}\mathbb{p} - \partial_{\mathbf{r}}(\gamma g(\alpha)\mathbf{r}\mathbb{p}) \\
&\quad - \partial_{r_\alpha}(\gamma_\alpha r_\alpha\mathbb{p}) \\
&= -\partial_\alpha(r_\alpha\mathbb{p}) - h(\boldsymbol{\theta}, \mathbf{r}, \alpha)\partial_{r_\alpha}\mathbb{p} \tag{18}
\end{aligned}$$

Inserting the coupling term $h(\boldsymbol{\theta}, \mathbf{r}, \alpha) = -\partial_\alpha g(\alpha)H(\boldsymbol{\theta}, \mathbf{r}) - \partial_\alpha \phi(\alpha)$ into the equation above, we can observe the Fokker-Planck operator vanishes. □

# B The Proof for Theorem 1

*Proof.* In a typical setting of numerical integration with associated stepsize $\eta$, one has

$$-\eta\nabla_{\boldsymbol{\theta}}\tilde{U}(\boldsymbol{\theta}) = -\eta\nabla_{\boldsymbol{\theta}}U(\boldsymbol{\theta}) + \sqrt{\eta}\mathcal{N}(\mathbf{0}, \eta\boldsymbol{\Sigma}(\boldsymbol{\theta})) \tag{19}$$

$$\eta\tilde{h}(\boldsymbol{\theta}, \mathbf{r}, \alpha) = \eta h(\boldsymbol{\theta}, \mathbf{r}, \alpha) + \sqrt{\eta\partial_\alpha g(\alpha)}\mathcal{N}(0, \eta\partial_\alpha g(\alpha)\sigma^2(\boldsymbol{\theta})), \tag{20}$$

which corresponds to the terms in SDEs

$$-\nabla_{\boldsymbol{\theta}}\tilde{U}(\boldsymbol{\theta})\mathrm{d}t = -\nabla_{\boldsymbol{\theta}}U(\boldsymbol{\theta})\mathrm{d}t + \sqrt{\eta\boldsymbol{\Sigma}(\boldsymbol{\theta})}\mathrm{d}\mathbf{W} \tag{21}$$

$$\tilde{h}(\boldsymbol{\theta}, \mathbf{r}, \alpha)\mathrm{d}t = h(\boldsymbol{\theta}, \mathbf{r}, \alpha)\mathrm{d}t + \sqrt{\eta\partial_\alpha g(\alpha)\sigma^2(\boldsymbol{\theta})}\mathrm{d}W_\alpha. \tag{22}$$

Then we derive the Fokker-Planck equation corresponding to the dynamics in Eq. (9) is

$$\begin{aligned}
\partial_t\mathbb{p}(\boldsymbol{\theta}, \mathbf{r}, \alpha, r_\alpha; t) &= -\partial_{\boldsymbol{\theta}}(\mathbf{r}\mathbb{p}) + \partial_{\mathbf{r}}\left(\nabla_{\boldsymbol{\theta}}U(\boldsymbol{\theta})\mathbb{p} + \gamma\mathbf{r}\mathbb{p}\right)\mathbb{p} + \frac{\eta}{2}\partial_{\mathbf{r}}\left(\boldsymbol{\Sigma}(\boldsymbol{\theta})\partial_{\mathbf{r}}\mathbb{p}\right) - \partial_\alpha(r_\alpha\mathbb{p}) \\
&\quad + \partial_{r_\alpha}\left(\gamma_\alpha r_\alpha\mathbb{p} - h(\boldsymbol{\theta}, \mathbf{r}, \alpha)\mathbb{p}\right) + \frac{\eta}{2}\partial_{r_\alpha}\left(\partial_\alpha g(\alpha)\hat{\sigma}^2(\boldsymbol{\theta})\partial_{r_\alpha}\mathbb{p}\right) + \partial_{\mathbf{r}}\left(\gamma\tilde{\beta}^{-1}(\alpha)\partial_{\mathbf{r}}\mathbb{p}\right) \\
&\quad - \frac{\eta}{2}\partial_{\mathbf{r}}\left(\boldsymbol{\Sigma}(\boldsymbol{\theta})\partial_{\mathbf{r}}\mathbb{p}\right) + \partial_{r_\alpha}\left(\gamma_\alpha\beta^{-1}\partial_{r_\alpha}\mathbb{p}\right) - \frac{\eta}{2}\partial_{r_\alpha}\left(\partial_\alpha g(\alpha)\hat{\sigma}^2(\boldsymbol{\theta})\partial_{r_\alpha}\mathbb{p}\right) \\
&= -\partial_{\boldsymbol{\theta}}\partial_{\mathbf{r}}\mathbb{p} + \partial_{\mathbf{r}}\partial_{\boldsymbol{\theta}}\mathbb{p} + \partial_{\mathbf{r}}(\gamma\mathbf{r}\mathbb{p}) - \partial_\alpha(r_\alpha\mathbb{p}) + \partial_{r_\alpha}(\gamma_\alpha r_\alpha\mathbb{p}) - h(\boldsymbol{\theta}, \mathbf{r}, \alpha)\partial_{r_\alpha}\mathbb{p} \\
&\quad - \partial_{\mathbf{r}}\left(\gamma\tilde{\beta}^{-1}(\alpha)\beta g(\alpha)\mathbf{r}\mathbb{p}\right) - \partial_{r_\alpha}\left(\gamma_\alpha\beta^{-1}\beta r_\alpha\mathbb{p}\right) \tag{23} \\
&= -\partial_\alpha(r_\alpha\mathbb{p}) - h(\boldsymbol{\theta}, \mathbf{r}, \alpha)\partial_{r_\alpha}\mathbb{p}. \tag{24}
\end{aligned}$$

Just plug $h(\boldsymbol{\theta}, \mathbf{r}, \alpha)$ into the Fokker-Planck equation to observe that it vanishes. □

# C Convergence Analysis

Since we apply stochastic approximation into CTLD, the convergence properties can be analyzed based on the SG-MCMC framework by [2].

Let $\boldsymbol{\theta}^*$ denote any local minima of $U(\boldsymbol{\theta})$ and its corresponding objective $U^*$, and $\{\boldsymbol{\theta}^{(1)}, \ldots, \boldsymbol{\theta}^{(L)}\}$ be a sequence of samples from the algorithm. The sample average can be defined as $\hat{U} = \frac{1}{L}\sum_{t=1}^{L} U(\boldsymbol{\theta}^{(t)})$. Our analysis focuses on using the sample average $\hat{U}$ as an approximation of $U^*$.

Denote the difference $\Delta U(\boldsymbol{\theta}) = U(\boldsymbol{\theta}) - U^*$ and the operators $\Delta B_t = \left(\tilde{U}(\boldsymbol{\theta}^{(t)}) - U\right)\nabla_{r_\alpha}$, $\Delta G_t = \left(\nabla_{\boldsymbol{\theta}}\tilde{U}(\boldsymbol{\theta}^{(t)}) - \nabla_{\boldsymbol{\theta}}U\right)^T\nabla_{\mathbf{r}}$. Under some necessary smoothness and boundedness assumptions (See Assumption 1 in the Supplementary Materials), we establish the following theorem to characterize the closeness between $\hat{U}$ and $U^*$ in terms of bias and mean square error (MSE). This also indicates the stability performance of our method.

**Theorem 2.** *The bias and MSE of $\hat{U}$ from CTLD with stochastic approximation w.r.t. $U^*$ are bounded with some positive constants $C_1$ and $C_2$,*

$$\left|\mathbb{E}[\hat{U}] - U^*\right| \le \frac{C_1 e^{-U^*}}{L} \sum_{t=1}^{L} \int e^{-\tilde{\beta}(\alpha^{(t)})\Delta U(\boldsymbol{\theta})} \mathrm{d}\boldsymbol{\theta} + C_2 \left( \frac{1}{L\eta} + \frac{\sum_t \mathbb{E}\left[\|\Delta G_t\| + \|\Delta B_t\|\right]}{L} \right) + \mathcal{O}(\eta)$$

$$\mathbb{E}(\hat{U} - U^*)^2 \le C_1^2 e^{-2U^*} \left( \frac{1}{L} \sum_{t=1}^{L} \int e^{-\tilde{\beta}(\alpha^{(t)})\Delta U(\boldsymbol{\theta})} \mathrm{d}\boldsymbol{\theta} \right)^2 + C_2^2 \left( \frac{1}{L\eta} + \frac{\sum_t \mathbb{E}\left[\|\Delta G_t\|^2 + \|\Delta B_t\|^2\right]}{L^2} \right)$$
$$+ \mathcal{O}(\eta^2)$$

Both of the two bounds involves two parts. The first one is the distance between the considered optima, $e^{-U^*}$ and the unnormalized annealing distribution, $e^{-\tilde{\beta}(\alpha^{(t)})\Delta U(\boldsymbol{\theta})}$, which is a bounded quantity related to $S$. The second part characterizes the approximation error introduced by stochastic approximation and numerical integration of SDEs.

Before presenting the proof for this theorem, we firstly present some necessary definitions and assumptions. We define a functional $\psi_t$ solving the following Poisson equation:

$$\mathcal{L}_t \psi_t(\boldsymbol{\theta}^{(t)}) = U(\boldsymbol{\theta}^{(t)}) - \bar{U}_{\tilde{\beta}_t}, \tag{25}$$

where $\mathcal{L}_t$ is the generator of the SDEs in Eq.(9) in the $t$-th iteration; and we define

$$\mathcal{L}_t f(\mathbf{y}_t) \triangleq \lim_{\eta \to 0^+} \frac{\mathbb{E}[f(\mathbf{y}_{t+\eta})] - f(\mathbf{y}_t)}{\eta}, \tag{26}$$

where $\mathbf{y}_t = (\boldsymbol{\theta}^{(t)}, \mathbf{r}^{(t)}, \alpha^{(t)}, r_\alpha^{(t)})$, and $f(\cdot)$ is a compactly supported twice differentiable function. The solution functional $\psi_t(\boldsymbol{\theta}^{(t)})$ characterizes the difference between $U(\boldsymbol{\theta}^{(t)})$ and the posterior average $\bar{U}_{\tilde{\beta}_t} = \int U(\boldsymbol{\theta}) \mathrm{p}_{\tilde{\beta}_t}(\boldsymbol{\theta}) \mathrm{d}\boldsymbol{\theta}$ for every $t$. Typically, Eq.(26) possesses a unique solution, which is at least as smooth as $U$ under the elliptic or hypoelliptic settings [18]. The function $\psi_t$ is assumed to be bounded and smooth:

**Assumption 1.** *$\psi_t$ and its up to 3rd-order derivatives, $\partial^k \psi_t$, are bounded by a function $\mathcal{V}(\mathbf{y}_t)$, i.e., $\|\partial^k \psi\| \le D_k \mathcal{V}^{p_k}$ for $k = 0, 1, 2, 3$, $D_k, p_k > 0$. Moreover, the expectation of $\mathcal{V}$ is bounded: $\sup_t \mathbb{E}\mathcal{V}^p(\mathbf{y}) < \infty$, and $\mathcal{V}$ is smooth such that $\sup_{s \in (0,1)} V^p(s\mathbf{x} + (1-s)\mathbf{x}') \le D(\mathcal{V}^p(\mathbf{x}) + \mathcal{V}^p(\mathbf{x}'))$, $\forall \mathbf{x}, \mathbf{x}', r \le \max\{2p_k\}$ for some $D > 0$.*

The proof for the bounded bias and MSE follows the framework proposed in [2].

*Proof. Bounded bias:*

Since we use the 1st-order integrator, we have

$$\mathbb{E}[\psi_t(\mathbf{y}_t)] = \tilde{P}_\eta \psi(\mathbf{y}_{t-1}) = e^{\eta \tilde{\mathcal{L}}_t} \psi(\mathbf{y}_t) + \mathcal{O}(\eta^2) = \left(\mathbb{I} + \eta \tilde{\mathcal{L}}_t\right) \psi_t(\mathbf{y}_{t-1}) + \mathcal{O}(\eta^2), \tag{27}$$

where $\eta$ is the step size of local numerical integrator, $\mathcal{L}_t$ is the generator of the SDEs ()-() for the $t$-th iteration, $P_h$ is its corresponding Kolmogorov operator, the $\tilde{\mathcal{L}}_t$ and $\tilde{P}_\eta$ represent the corresponding integrator and operator *with stochastic approximation*, respectively, and $\mathbb{I}$ denotes the identity map.

Sum over $t = 1, \ldots, L$ in Eq. (27), take expectation on both sides, and then inert the key relation $\tilde{\mathcal{L}}_t = \mathcal{L}_t + \Delta G_t + \Delta B_t$ to expand the first order term:

$$\sum_{t=1}^{L} \mathbb{E}[\psi(\mathbf{y}_t)] = \psi(\mathbf{y}_0) + \sum_{t=1}^{L-1} \mathbb{E}[\psi(\mathbf{y}_t)] + \eta \sum_{t=1}^{L} \mathbb{E}[\mathcal{L}_t \psi(\mathbf{y}_{t-1})] + \eta \sum_{t=1}^{L} \mathbb{E}[\Delta G_t \psi(\mathbf{y}_{t-1})]$$
$$+ \eta \sum_{t=1}^{L} \mathbb{E}[\Delta B_t \psi(\mathbf{y}_{t-1})] + \mathcal{O}(L\eta^2). \tag{28}$$

Now divide both sides by $L\eta$, utilize the Poisson equation (25) and rearrange all the terms, so that we obtain

$$\mathbb{E}[\frac{1}{L} \sum_t (U(\boldsymbol{\theta}_t) - U_{\beta_t})] = \frac{1}{L} \sum_{t=1}^{L} \mathbb{E}[\mathcal{L}_t \psi(\mathbf{y}_{t-1})] = \frac{1}{L\eta} \underbrace{(\mathbb{E}[\psi(\mathbf{y}_t)] - \psi(\mathbf{y}_0))}_{C_3}$$
$$- \frac{1}{L} \sum_t \mathbb{E}[(\Delta G_t + \Delta B_t)\psi(\mathbf{y}_{t-1})] + \mathcal{O}(\eta) \tag{29}$$

Then the bias can be bounded as follows,

$$
\left| \mathbb{E}\hat{U} - U^* \right| = \left| \mathbb{E}\left( \frac{1}{L}\sum_t (U(\boldsymbol{\theta}_t) - \bar{U}_{\beta_t}) \right) + \frac{1}{L}\sum_t \bar{U}_{\beta_t} - U^* \right|
$$

$$
\leq \left| \mathbb{E}\left( \frac{1}{L}\sum_t (U(\boldsymbol{\theta}_t) - U_{\beta_t}) \right) \right| + \left| \mathbb{E}\left( \frac{1}{L}\sum_t U_{\beta_t} - U^* \right) \right|
$$

$$
\leq C_1 U(\boldsymbol{\theta}^*)\left( \frac{1}{L}\sum_{t=1}^{L} \int_{\boldsymbol{\theta}\neq\boldsymbol{\theta}^*} e^{-\tilde{\beta}_t \hat{U}(\boldsymbol{\theta})}\mathrm{d}\boldsymbol{\theta} \right) + \left| \frac{C_3}{L\eta} \right| + \left| \frac{\sum_t \mathbb{E}[(\Delta G_t + \Delta B_t)\psi(\mathbf{y}_{t-1})]}{L} \right| + \mathcal{O}(\eta)
$$

$$
\leq C_1 U(\boldsymbol{\theta}^*)\left( \frac{1}{L}\sum_{t=1}^{L} \int_{\boldsymbol{\theta}\neq\boldsymbol{\theta}^*} e^{-\tilde{\beta}_t \hat{U}(\boldsymbol{\theta})}\mathrm{d}\boldsymbol{\theta} \right) + C_2\left( \frac{1}{L\eta} + \frac{\sum_t \mathbb{E}\left[\|\Delta G_t\| + \|\Delta B_t\|\right]}{L} \right) + \mathcal{O}(\eta),
$$

$$(30)$$

where the last inequality follows from the finiteness assumption of $\psi$, $\|\cdot\|$ represents the operator norm and can be bounded in the space of $\psi$ because of the assumption. These complete the proof for the bounded bias.

*Bounded MSE:*

The proof for the bounded MSE result is similar to that for the bounded bias. For the 1st-order integrator,

$$
\mathbb{E}[\psi_{\tilde{\beta}_t}(\mathbf{y}_t)] = (\mathbb{I} + \eta(\mathcal{L}_t + \Delta G_t + \Delta B_t))\psi_{\tilde{\beta}_t}(\mathbf{y}_{t-1}) + \mathcal{O}(\eta^2) \tag{31}
$$

Sum over $t$ from 1 to $L$ and insert the Poisson equation (25), divide both sides by $L\eta$ and then rearrange all the terms, we have

$$
\frac{1}{L}\sum_{t=1}^{L}(U(\boldsymbol{\theta}_t) - U_{\tilde{\beta}_t}) = \frac{1}{L\eta}(\mathbb{E}\psi_{\tilde{\beta}_L}(\mathbf{y}_{L\eta}) - \psi_{\tilde{\beta}_0}(\mathbf{y}_0)) - \frac{1}{L\eta}\sum_{t=1}^{L}(\mathbb{E}\psi_{\tilde{\beta}_{t-1}}(\mathbf{y}_{t-1}) - \psi_{\tilde{\beta}_{t-1}}(\mathbf{y}_{t-1}))
$$

$$
- \frac{1}{L}\sum_{t=1}^{L}(\Delta G_t + \Delta B_t)\psi_{\tilde{\beta}_{t-1}}(\mathbf{y}_{t-1}) + \mathcal{O}(\eta) \tag{32}
$$

Take the square of both sides, we can see that there exists some positive constant $C$ such that the following inequality holds.

$$
\left( \frac{1}{L}\sum_{t=1}^{L}(U(\boldsymbol{\theta}_t) - \bar{U}_{\tilde{\beta}_t}) \right)^2 \leq C\left( \underbrace{\frac{1}{L^2\eta^2}(\mathbb{E}\psi_{\tilde{\beta}_L}(\mathbf{y}_{L\eta}) - \psi_{\tilde{\beta}_0}(\mathbf{y}_0))^2}_{A_1} \right.
$$

$$
+ \underbrace{\frac{1}{L^2\eta^2}\sum_{t=1}^{L}(\mathbb{E}\psi_{\tilde{\beta}_{t-1}}(\mathbf{y}_{t-1}) - \psi_{\tilde{\beta}_{t-1}}(\mathbf{y}_{t-1}))^2}_{A_2}
$$

$$
\left. + \frac{1}{L^2}\sum_{t=1}^{L}(\Delta G_t^2 + \Delta B_t^2)\psi_{\tilde{\beta}_{t-1}}(\mathbf{y}_{t-1}) + \eta^2 \right)
$$

The term $A_1$ can be bounded by the assumption that $\|\psi\| \leq \mathcal{V}^{p_0} < \infty$. $A_2$ is bounded due to the fact that

$$
\mathbb{E}[\psi_{\tilde{\beta}_t}(\mathbf{y}_t)] - \psi_{\tilde{\beta}_t}(\mathbf{y}_t) \leq C_1\sqrt{\eta} + \mathcal{O}(\eta) \text{ for } C_1 \geq 0. \tag{33}
$$

This inequality holds since the the only difference between $\mathbb{E}[\psi_{\tilde{\beta}_t}(\mathbf{y}_t)]$ and $\psi_{\tilde{\beta}_t}(\mathbf{y}_t)$ lies in the additional Gaussian noise with variance $\eta$.

Now we have

$$
\mathbb{E}\left( \frac{1}{L}(U(\boldsymbol{\theta}_t) - \bar{U}_{\tilde{\beta}_t}) \right)^2 = \mathcal{O}\left( \frac{\sum_t \mathbb{E}[\|\Delta G_t\|^2 + \|\Delta B_t\|^2]}{L^2} + \frac{1}{L\eta} + \eta^2 \right) \tag{34}
$$

Finally, the MSE can be bounded as follows,

$$
\mathbb{E}\left( \hat{U} - U^* \right)^2 \leq \mathbb{E}\left( \frac{1}{L}\sum_t (U(\boldsymbol{\theta}_{t-1}) - \bar{U}_{\tilde{\beta}_t}) \right)^2 + \mathbb{E}\left( \frac{1}{L}\sum_{t=1}^{L}\bar{U}_{\tilde{\beta}_t} - U^* \right)
$$

$$
\leq CU(\boldsymbol{\theta}^*)^2\left( \frac{1}{L}\sum_{t=1}^{L}\int_{\boldsymbol{\theta}\neq\boldsymbol{\theta}^*} e^{-\tilde{\beta}_t\hat{U}(\theta)}\mathrm{d}\boldsymbol{\theta} \right)^2 + \mathcal{O}\left( \frac{\sum_t \mathbb{E}[\|\Delta G_t\|^2 + \|\Delta B_t\|^2]}{L^2} + \frac{1}{L\eta} + \eta^2 \right),
$$

$$(35)$$

which completes the proof for the bounded MSE.

$\square$

# D    Hyperparameter Settings

To facilitate the practical use of our method and reduce the number of hyperparameterss to be tuned, we always fix these parameters across all the experiments, $\sigma = 0.04$ and $K = 300$. The main parameters we need to tune are the learning rate and the momentum. In the following, we elaborate how other parameters are configured according to the learning rate.

**Friction Coefficients**    To set friction coeeficient $momentum$, the connection with SGD-Momentum provides us a direct guide for configuring the friction coefficients $\gamma$ and $\gamma_\alpha$ similar **as the momentum in SGD-Momentum**. Across all the experiments, we suggest this setting, $\gamma = (1 - c_m)/\eta$, where $c_m \in [0, 1]$ denotes the momentum coefficient to be tuned. For $\gamma_\alpha$, we set $\gamma_\alpha$ equal to $1/\eta$ corresponding to the momentum equal 0 to enable fast sampling across parameter space.

**Confining Potential Function**    To confine a reasonable temperature sampling range, we propose the configuration of $C$ as follows,

$$C = \delta'/\eta^2, \tag{36}$$

indicating the augmented variable $\alpha$ will be pulled to the origin once it touches the boundaries of the interval $[-\delta', \delta']$. This restricts the temperature to the desired range without loss of exploration abilities, while effectively avoiding the Hamiltonian system to spend too much time on sampling with high temperatures.

**Metadynamics**    The goal of metadynamics is to derive an asymptotically uniform distribution of the augmented variable $\alpha$ to achieve the transitions of between different modes of $\boldsymbol{\theta}$. Across the experiments, the Gaussian bandwidth $\sigma$ is set to be a constant 0.04. We divide the interval $[-\delta', \delta']$ into $K = 300$ parts. Empirical studies found that the proposed method is not sensitive to these parameters.

To control the convergence speed of metadynamics, we need to configure the value of Gaussian height $w$. According to Alg. 1, for metadynamics to take effects numerically, the magnitude of $w$ should be the same as:

$$w = \mathcal{O}(\frac{1}{\exp(-dst^2/2\sigma^2)\eta^2 L_s K}). \tag{37}$$

Where $dst$ is the length of sliced interval in the range $[-\delta', \delta']$ for metadynamics. The intuition behind this equation is that: in each update, the metadynamics would add a correction term $correct \curvearrowright w \exp(-dst^2/2\sigma^2)$ which would be computed $L_s K$ times in the exploration stage and considering the effects of learning rate $\eta$, the final magnitude of metadynamics correction on $r$ becomes: $correct \curvearrowright w \exp(-dst^2/2\sigma^2)\eta^2 L_s K$ which requires $w$ has similar magnitude of $\frac{1}{\exp(-dst^2/2\sigma^2)\eta^2 L_s K}$ to take effects. As the term $\exp(-dst^2/2\sigma^2)$ value is close to 1, and by multiplying 20 to enlarge the effects of metadynamics, we suggest the setting of $w$ as,

$$w = 20/(\eta^2 L_s K). \tag{38}$$

Thus, the proposed algorithm only needs the learning rate and the momentum to be adjusted that is almost as simple as SGD-Momentum. This will be shown in parameter settings section.

# E    Parameter Settings for Experiments

To ensure fairness for comparison, the additional parameters of newly proposed complex methods like AnnealSGD, Santa, ADAM and RMSprop are remained the same as their original paper. We do grid searches to find optimal values for each methods. Noted that the parameter searching for our proposed method is quite simple. Tuning the parameter of CTLD is quite simple and direct. For learning rate, we initially choose a learning rate which is the same according to its connection with SGD-Momentum and then decrease it gradually. Also according to its relationship with SGD-Momentum, we can derive a method to adjust CTLD's momentum like SGD-Momentum:

$$\gamma = (1 - c_m)/\eta, \tag{39}$$

where $c_m$ is the momentum coefficient to be tuned. Thus, tuning CTLD is almost as simple as SGD-Momentum. For $alpha$ dynamics momentum settings, we choose its momentum to equal 0 to enable the fast sampling across parameter space. So, the $\gamma_\alpha$ is:

$$\gamma_\alpha = 1/\eta. \tag{40}$$

### E.1 Stacked Denoising AutoEncoders

The batchsize is set as 128 and each layer is trained for $1 \times 10^5$ iterations across all experiments in this task. The momentum of the proposed CTLD is set to be 0 which is the same as SGD. $L_s$ is set to be $1.8 \times 10^4$. The learning rate is shown in Table. 1.

| METHOD | LEARNING RATE |
|---|---|
| SGD-M | 0.1 |
| RMSPROP | 0.001 |
| ADAM | 0.001 |
| ANNEALSGD | 0.1 |
| SANTA | 4E-11 |
| CTLD | 0.0008 |

Table 1: Learning Rate of SdAs

### E.2 Character Recurrent Neural Networks for Language Modelling

For our implementations, we referred to https://github.com/karpathy/char-rnn for initialization methods and model parameters. We used Wikipedia 100M dataset as it allowed us to pressure the learning and generalization ability of optimization methods. In this task, the batch size is set as 100 and all methods are used to train the model for 20 epochs. The $L_s$ is 9000 in this task. The momentum of our proposed method is 0.66. Although we have done very intensive grid search for SGD-M parameter search and even tried various factors for learning rate decreasing scheduler but the result for SGD keeps closely but still above 3.1 in training. The current best result is obtained when SGD's learning rate is 0.001 and momentum 0.9 with a factor scheduler every 5000 iterations and factor 0.85.

The learning rate is shown in Table. 2.

| METHOD | LEARNING RATE |
|---|---|
| SGD-M | 0.001 |
| RMSPROP | 0.002 |
| ADAM | 0.03 |
| ANNEALSGD | 0.0005 |
| SANTA | 8E-11 |
| CTLD | 2.08E-05 |

Table 2: Learning Rate of LSTM Neural Networks

## F  Computation time comparison

| Method | Computation Time(s) |
|---|---|
| SGD-M | 3.8471 |
| RMSprop | 4.1195 |
| Adam | 3.9437 |
| AdaDelta | 3.9327 |
| **CTLD** | **3.8868** |

Table 3: Average computation time of 1 epoch on mnist dataset with tensorflow backend for 10 epochs runs measured by python cProfile module. (Santa is a kind of adam with annealing noise)

Our current experiment implementations are based on MXNET 0.7 and lots of its operations are based on python making the implementations of 'adam' and 'rmsprop' significantly slower than them should be (https://github.com/dmlc/mxnet/issues/1516). Thus, we reimplement our method in Keras with tensorflow backend in mnist classification dataset. The testbed is a desktop computer with Intel I7 cpu and Nvidia Titan X GPU. Though more time is needed in keras to compile the computation graph, it can be observed that there is no significant overhead on our algorithm compared with other methods, which can be justified by the fact that our algorithm does not need to compute the power or sum of the large gradient matrix compared with RMSprop

and Adam. The largest overhead of our algorithm lies in the generation of random normal distribution variable which can be easily paralleled with mature APIs available within a modern GPU.

We used the Keras example implementation of MNIST cnn and measure the training time by the cumtime of training.py(-fit-loop).