[Reviews · NeurIPS 2017]

Reviewer 1



The paper proposes a method for improving stochastic gradient HMC. They add an extra term gaussian noise term to the momentum update controlled by a new parameter alpha. And they do metadynamics to more efficiently explore the landscape of this extra parameter. They fine-tune the solution at the end with a noise free momentum SGD. They show experimental results on stacked denoising autoencoders on MNIST and sequence modeling on wikipedia hutter prize. They show that their method outperforms state of the art methods on both datasets. evaluation : I think the paper is generally well written and the contribution is valuable as far as I can tell without being an absolute expert in this field. I have some doubts about the experiments which I detail below. I also have some doubts about the flat-sharp minimum argument and the relationship with the density lines 129-133. questions: - you rely on a two step process of bayesian optimization followed by momentum SGD. How important is the fine tuning ? How long does it have to run for ? - what is Ls in the experiments? Were both phases run in figure 2 or is the sgdm only run at the end ? - Why is SGD-M is missing form the figure 3 ? I came to think that well tuned momentum SGD matches performance in most state-of-the-art optimizers, judging by the tables in E1 and E2 the tuning involved is relatively basic. What are the results with 10 learning rates sampled on a log scale between 1e-4 and 1e-2 and a similar grid in momentum ? Are the best methods the same ? Can the proposed method beat that given the same computational budget ? - I think the problems may be too weak for the method to show its strength. I think imagenet or any of the experiments in [19] could make the case more convincing.

Reviewer 2



The paper proposes a new method to optimize deep neural networks, starting with a stochastic search using 'original' Langevin (where the temperature appears as a function of an auxiliary variable), then transitioning to more classical, deterministic algorithm. I enjoyed reading the paper - I am not an expert in the field but as far as I could tell the methods are novel, and the idea of treating the temperature as a function of an augmented variable seems elegant; since one can then change the landscape for temperature (tweaking g(\alpha) and \phi(\alpha)) without changing the optimum of the function. The numerical experiments seem to indicate that the method is not more computationally demand but improves optimization. I recommend acceptance, with minor caveats below. - The numerical experiments show that the algorithm obtain better loss than a number of classical neural network optimization methods. However they don't explicitly investigate the ability of the algorithm to jump between modes, a property frequently mentioned in the body of the text. Having an experiment on a toy, lower dimensional function, where that property could be clearly highlighted would have improved the paper. Generally speaking, visual representation of the behavior of the algorithm in a couple of canonical situation would have provided more intuition on the inner workings of the algorithm. The choice of networks (stacked DAE) and dataset (language modeling) seem also arbitrary. - Notation feels a bit clumsy. It is unfortunate to introduce an inverse temperature \beta, make it be equal to the output of a function g, and have it appear frequently as its inverse (i.e. the temperature). Why not introduce T(\alpha), \beta=1/T(\alpha) (just like Shannon's entropy does not require a k_B term, k_B does not need to appear here), and use either notation depending on which fits the equation most naturally (T for equations 4,9,10, beta for 6,7). It might also be arguably clearer to plot T(\alpha) in figure 1 (since the text talks about 'high temperature configuration' but the figure is showing low inverse-temperatures). - The paper suggests tweaking the value of \phi(\alpha) to obtain a desired distribution on \alpha. But is it not hard to understand what the marginal distribution of \alpha is, given the multiplicative term H(\theta,r) g(\alpha)? - Nitpicky: The force well from equation (11), as it is written, does not appear to have non-zero gradients. Does it not need to be smoothened to obtain the behavior described in the following lines? - In section 4.2, lines 206-211 are not particularly clear. Furthermore, wouldn't adding those gaussian kernel to the hamiltonian effectively act as 'walls' preventing alpha from mixing (transitioning from one side of the wall to the other seems to require going through a high energy configuration)? - Figure 2, \tilde beta is missing a backslash.

Reviewer 3



Disclaimer: I have never run Santa from [1] in practice, and might therefore have missed some key subtleties. The paper takes stochastic gradient Langevin dynamics, with Hamiltonian H(\theta, r), and basically makes the annealing schedule adaptive by introducing another parameter \alpha to the Hamiltonian H. \alpha adapts H with g(\alpha) * H and gets its own momentum variable. Via g(\alpha), the original Hamiltonian H can be suppressed, allowing the annealing schedule to be changed in the formulation. This set-up -- CTLD or continuously tempered Langevin dynamics -- is used in an algorithm to get parameters in a "flat minimum" area, after which standard SGD does some refinement. The results show that CTLD finds better solutions than Santa and methods that don't adaptively change the annealing schedule. CTLD adapts the temperature dynamics. A few pointers on the results: ** The paper added a prior or regularizer in line 72, which is (might be) required to let the posterior be normalizable and the dynamics simulate from the stationary distribution of the joint. However, in the results, the prior or a regularizer seems omitted? ** Why is drop-our required if there is already a large amount of extra 'exploration' noise introduced by CTLD? Is it fair to isolate a conclusion in the presence of this extra source of noise? For Hamiltonian MCMC for neural nets, why not go "all the way" as you already have samples (approximately) from the marginal p(\theta | data)? This is very much in the spirit of the original samplers for neural nets, e.g. [Neal, R. M. (1994) Bayesian Learning for Neural Networks, Ph.D. Thesis].